# Development and Validation of the “Program to Promote Positive Nursing Practice Environments”: Modified Delphi Study

**DOI:** 10.3390/healthcare12171748

**Published:** 2024-09-02

**Authors:** Soraia Pereira, Cintia Fassarella, Olga Ribeiro

**Affiliations:** 1Abel Salazar Biomedical Sciences Institute, University of Porto, 4050-313 Porto, Portugal; 2Northern Health School of the Portuguese Red Cross, 3720-126 Oliveira de Azeméis, Portugal; 3CINTESIS@RISE, 4050-313 Porto, Portugal; olgaribeiro@esenf.pt; 4Faculty of Nursing, University of Rio de Janeiro, Rio de Janeiro 20550-013, Brazil; cintiafassarella@gmail.com; 5Nursing School of Porto, 4200-450 Porto, Portugal

**Keywords:** work environment, nursing, Delphi study

## Abstract

Positive nursing practice environments benefit clients, professionals, and institutions, specifically for professionals, by improving professional visibility, recognition, retention intentions, well-being, commitment, job satisfaction, and reducing stress and burnout. Clients experience better quality care, safety, satisfaction, engagement, and fewer care omissions. Institutions favored reduced absenteeism and turnover, improved efficiency, lower costs, and better health outcomes. This study aimed to document the development of the “Program to Promote Positive Nursing Practice Environments” and analyze expert perceptions to reach a consensus. Using a two-round modified online Delphi technique with 22 experts, we examined perspectives on the program’s goals, duration, organization, and content. Initial feedback showed agreement on the program’s duration but mixed opinions on the number and length of sessions due to concerns about participant workload and fatigue. Adjustments led to a strong consensus and positive feedback on the program’s coverage of essential nursing practice environment constructs. The program includes sessions on client, professional, and institutional dynamics, focusing on key attributes and outcomes for fostering positive nursing environments. By equipping nurses with knowledge and skills, the program aims to enhance work environments effectively. These findings provide a substantial advancement in the field, offering a robust framework for future initiatives.

## 1. Introduction

Nursing practice environments have garnered increasing interest from researchers, such as Boudreau and Rhéaume (2024) and Sheen and Lee (2024), as well as organizations like the ICN and the WHO [1,2,3,4]. These entities have highlighted the significance of nursing practice environments in relation to the quality and safety of care. They emphasize the need to improve training and working conditions to not only ensure the satisfaction of professionals and the effectiveness of the care provided but also to enhance the sustainability of health systems [1,2,3,4]. The ICN’s 2024 International Nurses Day Report highlights that the health sector, particularly nursing, is crucial for the economic and social development of countries [1]. However, a substantial deficit of nursing professionals is projected in the coming years, with the WHO forecasting a shortfall of 18 million healthcare workers, including 4 million nurses, by 2030. This anticipated shortage is driven by an aging population and evolving disease patterns with a rise in chronic conditions such as diabetes and heart disease, which will increase the demand for specialized nursing care and escalating expectations for health services [1,2]. Infectious outbreaks, such as the COVID-19 pandemic, have exacerbated this situation. Following the initial enthusiasm and celebration of “frontline heroes”, conditions for these professionals have deteriorated, leading to high workloads, staff shortages, burnout, and low wages [1]. Addressing issues of overwork and underpayment is critical for ensuring the sustainability of health systems. The WHO emphasizes that fair wages and decent working conditions are essential not only for attracting and retaining professionals but also for acknowledging their invaluable contributions to society [2].

The ICN outlines five strategic priority areas for creating decent jobs for nurses: (1) investment in education and nurses through funding for nursing education and job creation; (2) valuing the work of nurses, including fair remuneration, recognition of their role, decent working conditions, and focus on nurses’ well-being; (3) gender equality in care work by reducing gender disparities and creating more equitable opportunities; (4) improving working conditions, which includes work-life balance, safe practice environments, and opportunities for professional development; and (5) collaborative work between the health sector and other sectors such as the economy, education, transport, and infrastructure to maximize the economic benefits of investing in nursing [1]. These strategic areas are crucial as they seek to enhance nurses’ working conditions, ensure they are properly valued and supported, and contribute to a more efficient and equitable healthcare system. Implementing these strategies can result in improved health outcomes, greater professional satisfaction, and a positive impact on global public health.

The “Nurse Investment, Prosperity, and Peace Chain” initiative illustrates that investing in nursing yields far-reaching benefits beyond healthcare. This initiative offers several critical benefits. Firstly, advancing nurse education and providing access to specialized training ensures that nurses acquire the latest knowledge and skills, leading to high-quality patient care and improved patient outcomes. Additionally, investment in nursing supports the creation of community health programs led by nurses, which effectively address local health needs. For example, community health nurses may lead vaccination campaigns, manage chronic diseases, or establish wellness clinics, directly enhancing community health. Furthermore, strengthening the nursing workforce enhances societal resilience. During emergencies, such as pandemics, a well-trained and supported nursing team can swiftly mobilize and respond, minimizing disruptions and contributing to societal stability. Additionally, effective investment in nursing supports healthier economies. By improving workforce productivity and reducing healthcare costs—such as through the prevention of burnout via supportive work environments and fair compensation—overall economic stability is bolstered. This, in turn, reduces turnover rates and the associated costs of hiring and training new staff. This holistic approach involves empowering nurses to fully utilize their skills, implementing funding models that support nurse-led services, developing an adequate workforce with diverse skills, ensuring universal access to nursing care, building robust educational and career progression pathways, fostering leadership roles among nurses, providing continuous professional growth opportunities, and supplying the necessary resources for optimal patient care [1].

The WHO also emphasizes the urgency of investing in nursing education, creating nursing jobs, and strengthening nursing leadership. The “State of the World’s Nursing 2020” report recommends improving working conditions by promoting professional safety, ensuring fair wages, respecting rights, and maintaining occupational safety [2].

These initiatives, by advancing nurse education, creating jobs with adequate working conditions, reducing inequalities, and fostering fair and safe work environments for nurses, are aligned with the United Nations Sustainable Development Goals (SDGs) [5]. They underscore the critical role of nursing in global sustainable development. The ICN1] and the WHO [6] stress the importance of enhancing nursing education, career progression, and continuous professional development opportunities, which directly support SDG 4, aimed at ensuring quality education and lifelong learning [5]. Moreover, creating decent jobs with fair wages and dignified working conditions for nurses aligns with SDG 8, as investing in nursing not only improves health outcomes but also strengthens the economy and fosters more resilient communities, promoting inclusive and sustainable economic growth [5]. The ICN’s focus on equitable opportunities and gender equality contributes to SDG 10 by advancing equality and combating gender discrimination [1,5]. The ICN’s “Nurse Investment, Prosperity, and Peace Chain” initiative illustrates that investment in nursing extends beyond health, contributing to stronger and more resilient societies [1]. Promoting safe work environments and upholding workers’ rights are consistent with SDG 16, which seeks to build effective and inclusive institutions and ensure peace and justice [5].

The connection between recommendations for improving nurses’ working conditions and the Quadruple Global Aim is also evident [5,7]. Investing in nursing education and working conditions enhances population health by ensuring that professionals are well-prepared and motivated to deliver high-quality care, addressing the nursing workforce shortage (SDG 3) [5,7]. Continuous training and professional development improve care quality, resulting in better patient experiences (SDG 4) [5,7]. Valuing nurses’ work and creating decent employment contribute to sustainable economic growth and reduce costs associated with turnover and absenteeism, thus lowering per capita healthcare costs (SDG 8) [5,7]. Additionally, promoting gender equality and equitable opportunities in nursing enhances job satisfaction and reduces burnout, creating a fairer and more rewarding practice environment for nurses (SDG 10 and SDG 16) [5,7]. Investment in nursing fosters more resilient and just societies, enhances the experience of healthcare providers, and aligns with the objectives of the Quadruple Global Aim, which aims to improve population health, the patient care experience, reduce costs, and enhance the experience of healthcare professionals [1,5,7].

Research on nursing practice environments has provided a crucial foundation for documenting the conditions under which nurses work, supplying essential data for the effective implementation of these initiatives. In fact, research on nursing practice environments and their relationship with outcomes for clients, professionals, and institutions has demonstrated a very clear impact [3,4,8,9,10,11,12]. Professionals working in positive nursing practice environments perceive greater visibility of nursing and the work of nurses [13]. They exhibit higher levels of intention to remain in their institutions and profession [14,15] and report greater well-being, commitment, and work-related quality of life [4,9,16,17,18,19]. They also experience higher job satisfaction, increased motivation, and lower levels of stress, emotional exhaustion, and burnout [3,4,10,20,21,22,23]. Clients perceive improvements in the quality of care provided [23,24,25], greater safety, and higher satisfaction with the care received [18,25,26]. They also report higher engagement and fewer instances of omitted care [4,27,28]. Institutions benefit from lower levels of absenteeism and staff turnover [19,24,29], resulting in improved efficiency and reduced costs [19,30]. Additionally, there are better indicators related to complication prevention and health gains [8,14,24].

Considering the aforementioned, and within the scope of a broader project on nursing practice environments and patient safety in primary healthcare, it was decided to develop a program aimed at promoting positive nursing practice environments and to validate it with an expert panel, capturing the experts’ perceptions of its suitability. The objective of this study is to document the process of developing the intervention program, analyze experts’ perceptions, and reach a consensus for the “Program for Promoting Positive Nursing Practice Environments”. Specifically, the study aimed to examine experts’ perspectives on the program’s goals, duration, organization, and content, using the Delphi technique. The key research question addressed was: “Are the objectives, duration, organization, and content of the “Program to Promote Positive Nursing Practice Environments” deemed adequate by the experts”?

## 2. Materials and Methods

### 2.1. Design

A descriptive design utilizing a modified Delphi technique was employed in this study. The initial version of the “Program to Promote Positive Nursing Practice Environments” is based on the analysis of the concept of a positive nursing practice environment, which was derived from a comprehensive scoping review on the topic. The scoping review was conducted in March 2023, following the methodology proposed by JBI for scoping reviews [31]. A comprehensive search strategy was employed across multiple databases, including MedLine, CINAHL, MedicLatina, and grey literature sources such as the Open Access Scientific Repository of Portugal and WorldCat. Publications were included with no language restrictions, provided they were published from 2007 onwards, the year when the ICN first introduced the concept of a “positive work environment”. This search yielded 166 studies, which facilitated the identification of the antecedents, attributes, and consequences of a positive nursing practice environment. These findings provided the foundational basis for the development of the program.

Therefore, the Delphi technique was instrumental in achieving consensus among 22 experts on the “Program to Promote Positive Nursing Practice Environments” [32]. Consensus was reached after a two-round Delphi survey. This study adhered to the recommendations for the Conducting and Reporting of Delphi Studies (CREDES) [33]. The study procedures, outlined in Figure 1, include: preparatory work with a scoping review, Survey Round I with online expert questionnaires, data analysis, Survey Round II with additional expert questionnaires, consensus-building, and finalizing the Program for Promoting Positive Nursing Practice Environments.

### 2.2. Selection of Experts

In this study, “experts” refers to professionals with extensive knowledge and experience in nursing practice environments or those working in primary health care. The inclusion criteria were as follows: (a) being a nurse for at least 10 years; (b) having experience/knowledge or conducting research in the area of nursing practice environments or working as a nurse in primary health care; and (c) having availability and interest in completing all the Delphi rounds. A total of 22 experts were included, consisting of nurses, specialist nurses, nurse managers, as well as nursing professors conducting research in this area. To broaden and diversify the perspectives obtained, participants from various regions were selected, provided they met the inclusion criteria. The questionnaire was pre-tested with nurses and researchers who met the inclusion criteria, and participants in the pre-test were excluded from taking part in the main study. This allowed for an analysis of the comprehensibility of all terms and questions posed.

Based on the previously defined criteria, participants were selected and recruited by sending emails to individuals within the researchers’ network, specifically targeting those who are or have been involved in the Positive Professional Environments for Nursing Practice (PPE4NursingPractice) project, a component of the CINTESIS@RISE host unit. The email included a formal invitation letter and a link to access the study. Upon accessing the study, participants were presented with information about the study and informed consent, which they had to accept before starting the questionnaire. Anonymity and confidentiality were guaranteed, and all participants were informed of the possibility of withdrawing from the research at any time without any repercussions.

### 2.3. Definition of Consensus

The fundamental premise of the Delphi technique is that consensus achieved by a group is more reliable than individual viewpoints [34,35]. Therefore, the Delphi technique is valuable for reaching consensus among experts in the field under study—nursing practice environments—or among nurses working in the context where the intervention program will initially be implemented, in this case, primary health care [32]. Consensus was defined using the classical percentage of agreement, predetermined based on scores on a 4-point Likert scale [36]. Specifically, consensus to revise an element was defined as an average importance score of 3.49 or less, while consensus to include the element without changes was defined as an average importance score of 3.5 or more. Also, questions with a coefficient of variation (CV) greater than 0.150 underwent analysis, and based on these values, were revised. The number of rounds required was determined by the level of consensus achieved [32].

### 2.4. Survey Round I

The questionnaires were developed using Microsoft Forms for online surveys. The questionnaire was divided into two sections: the first section consisted of questions regarding the sociodemographic and professional characteristics of the participants, and the second section focused on the “Program to Promote Positive Nursing Practice Environments.” In the second section, experts were asked to provide feedback on the program’s duration, session structure, and the length of each session. They were also queried on the program’s organization and the content to be covered in each proposed session. For questions rated as “disagree” or “strongly disagree”, experts were encouraged to suggest improvements. At the end of the questionnaire, all experts had the opportunity to provide general suggestions for the program. Questions with a coefficient of variation (CV) greater than 0.150 underwent analysis and were revised based on these values. All comments were analyzed and contributed to the program’s refinement. A reminder was sent to experts via email or phone message if they did not return the questionnaires within one week to ensure completion of the study.

### 2.5. Survey Round II

In the second round, experts were provided with a revised version of the program, incorporating feedback from the first round. To enhance accessibility of the intervention program questionnaire and organization, a schema was developed outlining session details, revised durations, and topics to be addressed in each session. The organization of time within each session was clarified to include breaks, as recommended by experts. For questions where experts indicated “Disagree” or “Strongly disagree”, suggestions for improvement were requested. At the end of the questionnaire, all experts were invited to provide general suggestions for the program. Scores and comments were collected and analyzed to determine expert consensus and finalize the structure, organization, and content of the “Program to Promote Positive Nursing Practice Environments”.

### 2.6. Data Analysis

The mean scores, percentage distributions of each rating, and the coefficient of variation (CV) for each item were thoroughly examined to determine the extent of consensus achieved among the expert panel. To support this analysis, demographic data and responses from each survey round were systematically exported into the Statistical Package for the Social Sciences (SPSS), version 27.0. A comprehensive descriptive analysis was then conducted on these datasets. Demographic data were analyzed to provide a clear profile of the expert panel, while the data from each round were separately assessed. This process led to the creation of a frequency table that detailed the experts’ responses, offering a structured and transparent representation of their feedback [32].

### 2.7. Ethical Considerations

The study was formally approved by the Ethical Committee of the Nursing School of Porto (approval number: CE_08/2024), ensuring compliance with ethical standards for research involving human participants. Prior to participation, each individual was thoroughly briefed on the study’s objectives, purpose, and the voluntary nature of their involvement. Participants were informed that they could withdraw from the study at any time without facing any negative consequences. All participants provided informed consent after being fully informed of their rights. This included assurances of anonymity in data processing, ensuring that personal information would not be identifiable in any reports or publications. The right to access information was emphasized, giving participants full transparency about the study and their role in it.

### 2.8. Rigour

The Delphi process was conducted according to the CREDES (Conducting and Reporting of Delphi Studies) guidelines [33]. This study was preceded by a comprehensive scoping review that resulted in an analysis of the concept of “positive nursing practice environment”, providing the foundation for organizing and including intervention program content. A pre-test round was conducted beforehand to address ambiguities, and questions were revised as necessary to establish research reliability [32]. Transparency in prior decisions regarding the selection of expert panelists, data collection, and consensus criteria enhanced the credibility of this study [37]. Inclusion criteria for experts were based on clinical experience or relevant knowledge of the study topic, encompassing not only nurses but also specialist nurses, nurse managers, and nursing professors with published work in the field. Formal invitations included detailed information about the study, its objectives, the importance of participation, and the potential benefits. All experts participated voluntarily, and collected information was anonymized and treated confidentially, mitigating group bias due to panel member anonymity.

The panel of 22 experts contributed to diversity of opinions and perspectives, supported by demographic information [38]. The number of experts ensured that no single opinion dominated the group. A low dropout rate from the first to the second round indicated panel consistency, stability, participant interest in the topic, commitment to the work, and study validity [39]. A total of two Delphi rounds were conducted, including open-ended questions for experts to provide comments and suggestions [40].

## 3. Results

### 3.1. Response Rate

To assess the effectiveness and engagement of the expert panel throughout the study, we meticulously tracked the response rates across the different rounds of questionnaires. This process ensured that the data collected were both comprehensive and reliable.

In the first round of questionnaires, 22 experts were contacted and invited to participate in the study. All invited experts agreed to participate and responded to the questionnaire, achieving a commendable response rate of 100%.

In the second round, the response rate remained high at 95%, with only one expert failing to respond within the specified timeframe, surpassing the literature’s recommended minimum of 70% [32]. Figure 2 provides a detailed depiction of the recruitment process, clearly outlining the steps involved in selecting and engaging the expert panel. Additionally, it details the response counts for each questionnaire round, offering insight into participant involvement at various stages. The figure also specifies the number of participants who completed each questionnaire, highlighting the level of expert engagement throughout the study.

### 3.2. Demographics

To ensure a thorough understanding of the expert panel’s composition and qualifications, we provide detailed demographic information about its members. These data highlight the breadth of experience and the academic and professional backgrounds that contributed to the study’s findings.

The expert panel comprised 22 specialists with an average of 21.3 years of professional experience. More than half of the experts hold a master’s degree or doctorate, indicating their high level of expertise. Regarding the distribution of the sample between the two countries of origin, 19 experts were from Portugal and 3 were from Brazil. In terms of job titles, the majority of the experts were engaged in clinical practice (n = 19) and worked in primary health care settings (n = 12). Table 1 provides detailed demographic information about the participants.

### 3.3. Survey Round I

In the first round of questionnaires, experts were asked about the program’s duration, session structure, length of each session, organization, and the content to be covered in each proposed session. While 95.45% (n = 21) agreed or strongly agreed with the overall program duration, agreement on the number and length of sessions dropped to 86.36% (n = 19). In response to open-ended questions, experts suggested that lengthy programs with numerous or extended sessions could lead to participant dropout due to difficulties in managing work during professional absences. They also noted that long sessions might be less productive and could cause participant fatigue. Given these suggestions and considering that these aspects received the lowest agreement among the experts, the program’s structure was reviewed, and a new proposal was presented in the second round of questionnaires.

Regarding the organization and content to be developed throughout the program, we achieved an average agreement mean score of 3.61 ± 0.49. The session with the lowest level of agreement was the first session (mean score 3.57 ± 0.50), which aimed to present the program’s objectives, methodology, strategies, content organization, and overall relevance. Conversely, session number 7, which aimed to address the outcomes of a positive nursing practice environment related to professionals, received the highest level of agreement (mean score 3.65 *±* 0.48). Table 2 presents the details of expert agreement scores and the Content Validity Index (CVI) from Survey Round I. It includes the agreement scores among experts for each item and the corresponding CVI, reflecting the extent to which the items were deemed relevant and valid by the experts.

Considering the levels of agreement obtained for each session and the issues identified with the program’s duration, number of sessions, and session length, the program was redesigned to incorporate the collected suggestions.

### 3.4. Survey Round II

To evaluate the effectiveness and acceptance of the revised program, a detailed analysis was conducted based on the second round of questionnaires. This analysis assessed the level of consensus and the quality of feedback provided by the participants.

The average agreement score of participants in the second round of the questionnaire achieved a consensus of 3.70 *±* 0.46, with responses predominantly indicating “strongly agree”. This was accompanied by a coefficient of variation (CV) of 0.125, the lowest variation observed across both rounds, and a scale-level content validity index based on the average method (S-CVI/Ave) of 1 [41].

Participants provided numerous valuable comments and suggestions, which will be incorporated into the session planning. These suggestions include integrating practical exercises in each session to promote interaction and addressing the influence of leadership on nursing practice environments.

The program received highly positive feedback from participants. One participant noted that the content was very interesting and that all constructs evaluated by existing instruments for assessing nursing practice environments were covered in the program. This program is designed to empower nurses with a comprehensive understanding of positive nursing practice environments and to equip them with the tools necessary to implement changes in their work environments. The final content and organization of the program were detailed in Table 3.

Briefly, Session 1 will delve into the characteristics and variables affecting both clients and professionals. It will cover aspects such as the professionalism and clinical expertise of nurses, effective leadership practices, and the critical role of health literacy for clients. Additionally, the session will highlight the importance of timely access to healthcare services. Session 2 will focus on the institutional dimensions of nursing practice. This includes exploring the safety culture within healthcare settings, various models of clinical supervision in nursing, strategic planning for nursing, and methodologies for organizing work within institutions. Session 3 will address the key attributes of a positive nursing practice environment. Discussions will center on the significance of teamwork and communication, the role of engagement and shared decision-making, and the necessity of having clearly defined nursing fundamentals. Session 4 will examine the positive outcomes of an effective nursing practice environment. It will cover improvements in client-related areas such as patient safety, client satisfaction, and engagement. Furthermore, it will address institutional benefits like reduced absenteeism and enhanced health indicators. Session 5 will explore the impact of a positive nursing practice environment on professionals. Key outcomes will include enhanced professional well-being, reduced stress and burnout, and improvements in job quality of life. The session will consider the wide range of benefits that a supportive work environment can offer to nurses. Session 6 will involve a group activity designed to synthesize and consolidate the content covered throughout the program. The session will also include a conclusion, where participants will provide feedback on the program and discuss suggestions for improvement.

## 4. Discussion

This study aimed to document the detailed process of developing the intervention program, validate it through a comprehensive analysis of experts’ perceptions, and achieve consensus on the “Program for Promoting Positive Nursing Practice Environments”. The development process involved several stages, including initial concept formulation, iterative feedback collection, and program refinement based on expert input. The modified Delphi technique, characterized by its iterative feedback rounds and expert consensus building, proved to be an effective method for understanding the experts’ perspectives on the program’s suitability. Utilizing an online format allowed us to easily access specialists from various regions and to incorporate their insights, enriching the feedback process [35]. The first round involved a detailed questionnaire that set the groundwork for expert opinions, while the subsequent round focused on refining the program based on initial feedback. Although the first round was more extensive than the second, the response rate exceeded our expectations. This indicates that, despite its length, the questionnaire was easy to complete and that the experts found the topic and work to be meaningful.

The duration of the program emerged as a critical factor during development. Balancing comprehensiveness with practical time constraints was essential to ensure that each session adequately covered key topics while remaining manageable for participants. The program spans a total of 18 h, organized into six sessions of three hours each, designed to deliver a thorough and practical exploration of the positive nursing practice environment. The content is strategically structured to address influencing factors, characteristics, and outcomes associated with positive work environments. This comprehensive approach is intended to provide nurses with actionable insights and practical tools to enhance their work settings effectively.

Aiming to be comprehensive, the program content encompasses all aspects that influence, characterize, and result from a positive nursing practice environment. It also seeks to equip nurses with the knowledge and skills necessary to promote and enhance their work environment. The “Program to Promote Positive Nursing Practice Environments” consists of sessions covering topics such as characteristics and variables related to clients, professionals, and institutions, attributes of a positive nursing practice environment, and the outcomes such an environment can produce for clients, professionals, and institutions. This program could represent a significant asset in promoting a positive nursing practice environment by integrating various aspects related to clients, professionals, and institutions, as well as the attributes of a nursing practice environment.

Encouraging nurses’ involvement in the management and participatory structures of nursing practice environments may be a compelling strategy to empower them to voice their opinions and influence decisions affecting their daily routines [9]. Effective leadership skills and strategies that promote a positive nursing practice environment, such as participative management and supportive organizational practices, can profoundly drive these changes. According to a study by Amaliyah and Tukimin [42], interventions focusing on workplace improvement should prioritize managerial and organizational support, along with access to resources and opportunities. This aligns with Paguio et al. [25], which underscores that a participatory approach provides a foundation for enhancing nurses’ work environments, promoting their autonomy, satisfaction, and care quality.

Kol et al. [43] also support this perspective, stating that strategies fostering positive work environments include participative management structures, foundational nursing principles for quality care, support for educational/professional development and autonomy, staffing adequacy, and collegial relationships among professionals. These strategies are widely recognized through Magnet certification. Magnet certification is highlighted in various studies as promoting better work environments through structural empowerment of professionals, transformational leadership policies, exemplary evidence-based practice, innovation, and improvement [44].

Providing support for professional development and promoting nurses’ autonomy also appears crucial. Specific sessions are designed to equip nurses with advanced techniques for active participation in continuous workplace improvements. These sessions focus on fostering leadership skills, enhancing problem-solving capabilities, and encouraging proactive involvement in organizational changes. Paguio et al. [25] identify this as an effective intervention in promoting positive nursing practice environments through quality improvement processes and action planning, emphasizing nurse autonomy resulting from a participatory approach. Donley [19] also highlights the critical role of autonomy and its promotion, noting that professionals with high levels of autonomy and influence over work decisions report higher levels of personal fulfillment [10].

Fostering an organizational culture that supports nurses is essential. This program aims to provide strategies for nurses to actively shape and positively influence their organizational culture. This includes fostering collaborative environments, recognizing and addressing cultural barriers, and involving nurses in initiatives that promote a positive and rewarding work environment. The importance of organizational culture is underscored as a significant priority for nurses, highlighting its critical role in the effectiveness of interventions targeting nursing practice environments, as noted by Paguio et al. [25] and Azevedo et al. [45]. Therefore, addressing organizational culture and motivating nurses to engage in organizational activities and projects are essential steps. A positive organizational culture based on support and collaboration among professionals is foundational to a positive work environment [46,47].

Integrating strategies for nurses’ self-care and well-being into this program, considering professionals as a strategic pillar, could also contribute to helping nurses better cope with stress or workloads more effectively. Addressing nurses’ self-care and well-being needs is essential for promoting nurses’ performance and productivity. Promoting resilience by providing resources that foster a positive attitude and motivating nurses to be agents of change and promoters of positive nursing practice environments are critical aspects [46].

Stand-alone strategies such as implementing work schedules that improve work-life balance, stimulating younger nurses through participation in projects, promoting generational diversity within teams [44], workload monitoring, task redistribution based on team characteristics, strategies to promote self-control [48], and rituals promoting team cohesion have been documented in recent literature [8,49,50,51]. Developing programs that integrate multiple strategies and focus on empowering professionals with tools to improve their work environment is not common and is often associated with magnet certification, beacon recognition, or HWE standards [28,43,44,50].

The proposed intervention program focuses on clients, professionals, and institutions. Its goal is to empower nurses, enhance their skills and knowledge, and implement positive changes in nursing practice environments. This improvement not only benefits nurses individually but also enhances care quality, fostering a more satisfying and sustainable work environment for all.

The results of this study represent a significant advancement in this field and serve as a foundation for the design of other programs focused on promoting positive nursing practice environments. The development of programs that foster positive nursing practice environments is aligned with international priority agendas and the Sustainable Development Goals (SDGs) [5]. Such programs address critical aspects of healthcare quality, professional well-being, and equitable work environments for nurses. Specifically, they support SDG 3 (Good Health and Well-being) by enhancing care quality and mitigating burnout among healthcare professionals. They also contribute to SDG 4 (Quality Education) by advancing educational and professional development for nurses and to SDG 8 (Decent Work and Economic Growth) by promoting fair working conditions and improved job satisfaction. These contributions highlight the program’s significant role in advancing global health priorities and supporting sustainable development. Numerous studies have demonstrated the need for and benefits of improving nursing practice environments. However, for these changes to be effective, it is essential to involve individuals—in this case, nurses—to act as agents of change and leaders in promoting positive nursing practice environments.

This study incorporated a purposively selected panel of experts due to time and financial constraints, which may have led to the exclusion of some potential experts in this field. Nevertheless, the research team aimed to ensure a comprehensive and diverse panel with an adequate number of participants to encompass varied perspectives. These limitations should be acknowledged as they may affect the generalizability of the findings. While the robustness of the developed program was considered, further studies are necessary to validate and evaluate its reliability in practice.

## 5. Conclusions

This study developed and validated the “Program for Promoting Positive Nursing Practice Environments” through an extensive Delphi process. The program addresses critical aspects of nursing practice by focusing on characteristics and variables related to clients, professionals, and institutions, as well as the attributes and outcomes of a positive nursing practice environment. The final structure of the program, refined based on expert feedback, includes six sessions that cover essential topics: the factors influencing clients and professionals, institutional dimensions, attributes of a positive practice environment, and its impact. This structure provides a comprehensive approach, equipping nurses with the knowledge and skills necessary to improve their work environments. The high response rates and positive feedback from the expert panel affirm the program’s relevance and effectiveness. The insights gained from experts led to refinements in the program’s structure and content, ensuring that it addresses both theoretical and practical aspects of nursing practice environments thoroughly. Although the study’s results represent a significant advancement in the field and lay a foundation for similar programs, further research is needed to assess the program’s effectiveness in real-world settings.

In summary, this study highlights the essential role of nurses as change agents and leaders in fostering positive nursing practice environments. It underscores the importance of continued efforts to implement and expand programs aimed at enhancing nursing practice and patient care. The findings offer a valuable framework for initiatives focused on improving work environments and advancing care quality.

## Figures and Tables

**Figure 1 healthcare-12-01748-f001:**
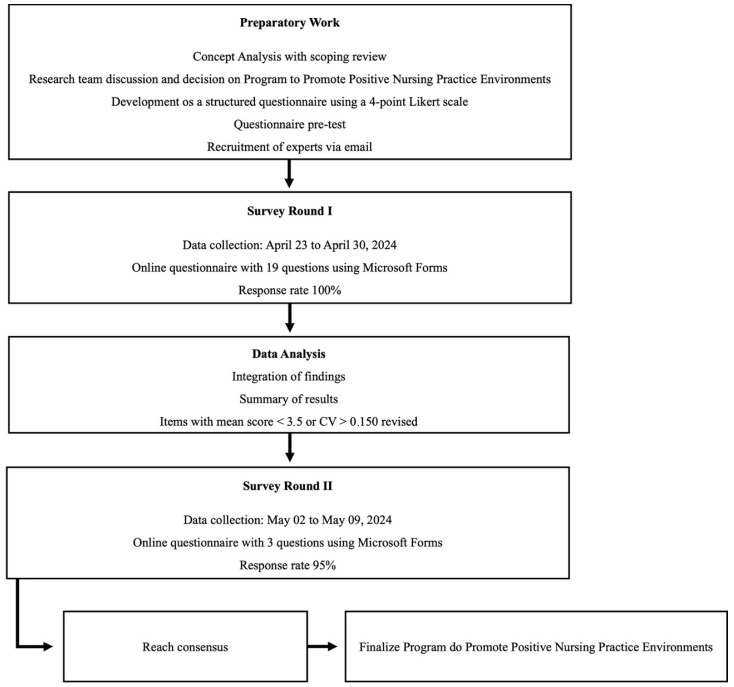
Flowchart with systematized stages of the Delphi process.

**Figure 2 healthcare-12-01748-f002:**
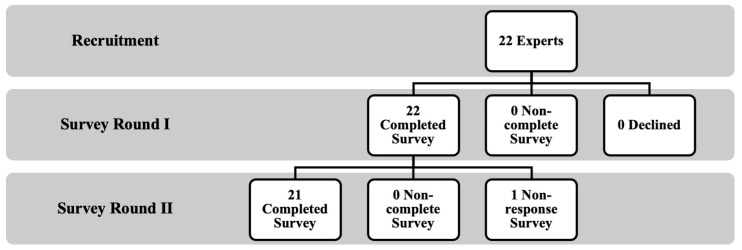
Recruitment and completion of the Delphi survey rounds.

**Table 1 healthcare-12-01748-t001:** Demographic information about the participants.

	I Round Survey (n = 22)		II Round Survey (n = 21)	
	Frequency	Percentage	Frequency	Percentage
**Gender**				
Female	17	77%	16	76%
Male	5	23%	5	24%
**Years**				
<40	6	27%	5	24%
40–50	13	59%	13	62%
>50	3	14%	3	14%
**Nationality**				
Portuguese	19	86%	19	90%
Brazilian	3	14%	2	10%
**Education**				
Graduate	8	36%	8	38%
Master’s	11	50%	10	48%
Doctorate	3	14%	3	14%
**Job Title**				
Nurse	1	5%	1	5%
Nurse Specialist	16	73%	15	71%
Nurse Manager	2	9%	2	10%
Professor	3	14%	3	14%
**Years of Experience**				
10–19	8	36%	7	33%
20–29	11	50%	11	52%
≥30	3	14%	3	14%
**Workplace**				
Primary Health Care	12	55%	11	52%
Hospital	7	32%	7	33%
Academic Institution	3	14%	3	14%

**Table 2 healthcare-12-01748-t002:** Details of expert agreement scores and content validity index in survey round I.

	Item	Mean (SD)	CV	I-CVI *
1	Total duration of the program	3.52 (0.58)	0.165	0.95
2	Number of sessions	3.39 (0.64)	0.189	0.90
3	Duration of each session	3.17 (0.70)	0.221	0.86
4	Content of Session 1	3.57 (0.50)	0.139	1
5	Content of Session 2	3.61 (0.49)	0.135	1
6	Contents of Session 3	3.61 (0.49)	0.135	1
7	Contents of Session 4 and 5	3.61 (0.49)	0.135	1
8	Contents of Session 6	3.61 (0.49)	0.135	1
9	Contents of Session 7	3.65 (0.48)	0.130	1
10	Contents of Session 8	3.61 (0.49)	0.135	1

* CV = coefficient of variation; I-CVI = item-level content validity index; SD = standard deviation.

**Table 3 healthcare-12-01748-t003:** Content and organization of the Program for Promoting Positive Nursing Practice Environments.

Session	Duration	Content
Session 1	3 h	Presentation of the Program to Promote Positive Nursing Practice Environments. Characteristics and variables related to clients and professionals.
Session 2	3 h	Characteristics and variables related to institutions.
Session 3	3 h	Attributes of a Positive Nursing Practice Environment.
Session 4	3 h	Outcomes of a Positive Nursing Practice Environment related to institutions and clients.
Session 5	3 h	Outcomes of a Positive Nursing Practice Environment related to professionals.
Session 6	3 h	Group dynamics. Conclusion of the Program to Promote Positive Nursing Practice Environments

## Data Availability

The datasets analyzed during the current study are available from the corresponding author upon reasonable request.

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
