# Peer review of "Development and Validation of the “Program to Promote Positive Nursing Practice Environments”: Modified Delphi Study"

_healthcare, 2024, doi:10.3390/healthcare12171748_

Round 1

Reviewer 1 Report

Comments and Suggestions for Authors

Methodologically well-written article. Concrete conceptual framework, based on current scientific evidence. Methodology that clarifies the study carried out. I felt the need to present more data, namely the content validity index by item and the overall content validity index. Interesting and topical article: congratulations!

Author Response

Comments: “Methodologically well-written article. Concrete conceptual framework, based on current scientific evidence. Methodology that clarifies the study carried out. I felt the need to present more data, namely the content validity index by item and the overall content validity index. Interesting and topical article: congratulations!”

Response: Dear Reviewer, thank you very much for your time and expertise in reviewing our work. We appreciate your valuable input and agree that including the Content Validity Index by item and the overall Content Validity Index would be a significant contribution. Accordingly, we have added a table on page 8, lines 305-306, detailing the Expert Agreement Scores and Content Validity Index. Additionally, we have included information on the scale-level Content Validity Index based on the average method in the results description of the second round. Thank you once again for your interest and thoughtful feedback!

Reviewer 2 Report

Comments and Suggestions for Authors

This is an interesting issue of discussion with an innovative approach that covers a significant problem with the shortage of nursing staff and the need for creating positive working environents. The design of the study, the abstract and introducation are fine. I would only recomend to the authors to try and be more specific. It is useful to start a paragraph with a thematic sentece that focuses to the point of what they study. There are many generalizations in the Introduction part but still the text is fine.

I would suggest improvements in the part of "material and methods" and specifically to refer analytically in the way they assessed their data. Which were the research tools and more information about them. 

Finally, it would be interesting to see an expanded version of the Discussions and the Conclusions with detailed analysis.

Overall, this is an article that worths publishing after the minor revisions mentioned above.

Very well done!

Author Response

Comment 1: “This is an interesting issue of discussion with an innovative approach that covers a significant problem with the shortage of nursing staff and the need for creating positive working environents. The design of the study, the abstract and introducation are fine. I would only recomend to the authors to try and be more specific. It is useful to start a paragraph with a thematic sentece that focuses to the point of what they study. There are many generalizations in the Introduction part but still the text is fine.”

Response 1: Dear Reviewer, thank you very much for your time and expertise in reviewing our work. We appreciate your valuable input and acknowledge that some parts of the introduction were somewhat generalized. We have made several revisions to enhance the writing, making it more pragmatic, fluid, and objective.

Comment 2: “I would suggest improvements in the part of "material and methods" and specifically to refer analytically in the way they assessed their data. Which were the research tools and more information about them.” 

Response 2: Your comments were crucial in improving our work, and we are grateful for them. In the Materials and Methods section, we have added details about the preparatory work, which included a comprehensive scoping review aimed at mapping the evidence and contributing to the analysis of the concept of a 'positive nursing practice environment.' The clearly detailed process and findings from this analysis, which form the basis of our program, are currently under peer review in another journal. We hope the revised description sufficiently addresses any questions regarding the research conducted.

Comment 3: “Finally, it would be interesting to see an expanded version of the Discussions and the Conclusions with detailed analysis.”

Response 3: Thank you again for your feedback. We have made some adjustments to the discussion to provide more detail and have rewritten the conclusion to better align with the work completed."

Reviewer 3 Report

Comments and Suggestions for Authors

It is an attempt to develop and analyze on program to promote nursing practice environments by the modified Delphi method. It is a meaningful research to refine the program based on feedback to ensure it effectively addresses critical aspects.

Thanks for the opportunity to peer review.

The purpose of the research is clear and consistent. Comments and suggestions are below as following

1. Please describe more about preparatory work for development structured questionnaire for "Program to Promote Positive Nursing Practice Environments" in the paragraph of the method and results.

2. Please create a table to describe the result of each session in survey round I.

3. Please check the correctness of the description about consensus to revise element consideration score and questions with a coefficient of variation underwent analysis, in paragraph of consensus definition (at lines 165-168) and survey round I (at line 181). The authors should revise to present it as points instead of commas.

4.Could the authors describe clearly each content of the program for promoting positive nursing practice environment in table 2, such as outcome of the nursing practice environment related clients, professionals and institutions?

Comments on the Quality of English Language

The manuscript requires minor editing.

Author Response

Comment 1: “Please describe more about preparatory work for development structured questionnaire for "Program to Promote Positive Nursing Practice Environments" in the paragraph of the method and results.”
Response 1: Dear Reviewer, thank you very much for your time and expertise in reviewing our work. We appreciate your valuable input. In the Materials and Methods section, we have added details about the preparatory work, which included a comprehensive scoping review aimed at mapping the evidence and contributing to the analysis of the concept of a 'positive nursing practice environment.' The clearly detailed process and findings from this analysis, which form the basis of our program, are currently under peer review in another journal. We hope the revised description sufficiently addresses any questions regarding the research conducted.

Comment 2: “Please create a table to describe the result of each session in survey round I.”
Response 2: Thank you! We have added a table on page 8, lines 305-306, detailing the Expert Agreement Scores and Content Validity Index in survey round I.

Comment 3: “Please check the correctness of the description about consensus to revise element consideration score and questions with a coefficient of variation underwent analysis, in paragraph of consensus definition (at lines 165-168) and survey round I (at line 181). The authors should revise to present it as points instead of commas.”
Response 3: Thank you very much for pointing out this oversight. We have made the corrections as mentioned and have updated the variance coefficient values to use periods instead of commas.

Comment 4: “Could the authors describe clearly each content of the program for promoting positive nursing practice environment in table 2, such as outcome of the nursing practice environment related clients, professionals and institutions?”
Response 4: We appreciate your feedback. We have included a brief description of each topic that will be covered in the sessions. As noted in our response to Comment 1, this content is derived from a concept analysis study that is currently under peer review in another journal. Consequently, we have provided a brief description in this document. Thank you again for your valuable comments.

We have also conducted a thorough linguistic review of the entire document to enhance its writing and readability.